# Household Factors of Foodborne Diarrhea in Children under Five in Two Districts of Maputo, Mozambique

**DOI:** 10.3390/ijerph192315600

**Published:** 2022-11-24

**Authors:** Nórgia Elsa Machava, Fhumulani Mavis Mulaudzi, Elsa Maria Salvador

**Affiliations:** 1Department of Nursing Science, Faculty of Health Sciences, University of Pretoria, Pretoria 0028, South Africa; 2Department of Biological Sciences, Faculty of Science, Eduardo Mondlane University, Maputo P.O. Box 257, Mozambique

**Keywords:** foodborne diarrhea, caregivers, children under five, factors

## Abstract

Household factors involved in the disease of diarrhea are multifaceted. This study aimed to explore and describe the household factors affecting foodborne diarrhea in children younger than 5 years old using structured questionnaire data based on quantitative tools. The sample size was calculated based on a binomial distribution. A total of 300 children, together with their caregivers, participated, and the data were descriptively and mathematically analyzed using Epi Info modelling. The caregivers were mostly female and included 93.3% rural and 84% urban dwellers of ages between 18 and 38, who were single but living with someone. Of the children who were under six months of age, 23.3% in rural areas and 16.6% in urban areas had diarrhea, while of the children between 12 and 23 months of age, 36.6% in urban areas and 30% in rural areas had diarrhea. The relatives had similar symptoms before the child became ill, with 12.6% of relatives in rural areas and 13.3% in urban areas reporting this. Before receiving medical assistance, 51.3% of children in rural areas and 16% of children in urban areas were treated with traditional medication. Water was not treated before drinking in 48% of rural cases and 45.3% of urban cases. A total of 24.6% of infants in urban areas and 12.6% of infants in rural areas used a bottle for feeding. The factors affecting foodborne diarrhea were the use of traditional medication in rural areas, bottle feeding in urban areas and untreated water used for drinking in both areas.

## 1. Introduction

Diarrhea is the most important foodborne communicable disease among children under five. It is the fifth main cause of death in children under five, and about 2.5 million children die annually in the world due to it [1,2]. Diarrhea is considered the third commonest cause of disease in children globally [3]. Diarrhea is more critical in developing countries, where it is considered the second commonest cause of death, and around 22% of childhood deaths are attributable to diarrhea [4].

In sub-Saharan Africa, the morbidity of diarrhea is associated with poverty and other socio-demographic factors [5]. Appropriate drinking water, food handlers, poor sanitation, deficient hygiene behaviors (WASH) and unclean environmental sanitary conditions are the principal factors that cause diarrhea [6,7,8]. Among these factors, insufficient WASH is the main factor. In total, 30% of people around the world do not use a safely managed drinking water service. In places where water services are available, in many cases, the water is contaminated. Approximately 2.3 billion people lack elementary sanitation services, and more than 60% do not consume appropriate water and dispose of their excreta in an inadequate way [9].

In Mozambique, 56% of households have adequate access to water, and 39% of households have access to improved sanitation [10]. Various studies on low- and middle-income countries (LMICs) have shown high levels of microbial contamination in children’s foods [11,12]. Children are quickly exposed to fecal material, unclean foods and other contaminated substances, such as water, human hands, soil and objects. Recurrent contact with contaminated materials can cause foodborne diarrhea in young children [13].

Among children under five, the age group between 6 and 11 months are most commonly affected by diarrhea because, in this age group, the children start to eat complementary foods, which can also be contaminated. Furthermore, this is the age when children come into contact with soil and test things by mouth [14].

There are many factors of foodborne diarrhea. There is a need to know these factors in order to intervene in three aspects of the disease, namely its prevention, treatment and control. The question of the present research was: what are the household factors of the origin of foodborne diarrhea in children younger than five? To address the question of the research, we explore and describe the household factors of foodborne diarrhea in children younger than five years of age.

## 2. Materials and Methods

The study took place in two districts of Maputo, namely the rural area known as the Marracuene district, with 230,530 inhabitants, of which 29,277.32 are children under five, and the urban area known as the Kamaxakeni district, with 203,660 inhabitants, of which 20,930 are children under five years of age. The data were collected using face-to-face interviews based on a structured questionnaire. The data collection questionnaire was piloted to reduce probable inaccuracy and eliminate possible incongruences. Caretakers attending the health facility with children with diarrhea infections were interviewed. The sample size was calculated based on a binomial distribution. A total of 300 children, together with their caregivers, participated in the study, with 150 children from urban areas and 150 children from rural areas. The calculations were based on an assumption of an average prevalence of less than 10%, assuming a sensitivity and specificity of 60% and 99%, respectively, and a precision of 5% at the 95% confidence level. The sample size calculations were conducted with Epitool (http://epitools.ausvet.com.au/content.php?page=home, accessed on 10 February 2020).

All the children under five with diarrhea infections at the health facility who were accompanied by their caregivers were required to have lived in the same household for at least 3 months. The data were analyzed based on descriptive and mathematical modelling (Epi Info). Prior to the study, the protocol was approved by the ethical committee of the Faculty of Health Science at the University of Pretoria, South Africa, with the registration number 595/2020-Line 1. The research was authorized by the health facility where the study took place. The confidentiality of all the data collected was secured, and anonymity was ensured by using codes to identify the participants.

## 3. Results

### 3.1. Socio-Demographic Characteristics

The children were mostly accompanied by their mothers (Table 1). The age and gender of the caregivers can be seen in Table 1. The marital status of the caregivers was single but living with someone for 76.6% (115) of individuals in rural areas and 68.6% (103) of individuals in urban areas and married for 18% (27) of individuals in urban areas and 14% (21) of individuals in rural areas. The age and gender distribution of the children with foodborne diarrhea can be seen in Table 1. Table 2 describes the occupations and monthly income of the caregivers. In the household, the income provider was male in 48.6% (73) of cases and female in 17.3% (26) of cases in rural areas and male in 57.3% (86) of cases and female in 26.6% (40) of cases in urban areas. A description of the rooms and number of individuals per room used for sleeping can be found in Table 3. The construction materials of the houses visited in the present study are described in Table 4.

### 3.2. Disease-Related Questions

Table 5 provides a description of the actions that the caregivers performed at the household level when the children became ill with diarrhea and the time taken before they received medical assistance. In addition, before medical assistance, the children were treated in the household with traditional medications, such as leaves and roots in 51.3% (77) of cases in rural areas and 16% (24) of cases in urban areas. Three months before the commencement of diarrhea, a few caregivers administered antibiotics to their children. These were given for the treatment of cough and diarrhea. The majority of children were vaccinated according to their age, following the Mozambican program of vaccination that includes the rotavirus vaccine at two and four months of age. Only 18.6% (28) of rural children and 19.3% (29) of urban children were not vaccinated. It was found that a small proportion of the children were admitted to hospital because of diarrhea, including only 2% (3) of rural children and 8.6% (13) of urban children. It was found that before the child became sick, there were other relatives with similar symptoms in the same household in 12.6% (19) of rural and 13.3% (20) of urban cases. After the sickness of the child, other individuals within the household became sick in 4.6% (7) of rural cases, and only one person was sick in the urban areas.

The majority of the caregivers reported having acquired food at the local market. The possible sources of diarrhea infection are described in Table 6, and the places where raw and cooked meat were acquired for the household are described in Table 7. Allergies to shellfish in the children were reported at a rate of 13.3% (40) in both districts.

In the rural area, about 73% (109) of children were infants, and 72% (108) were infants in the urban areas. Different types of water were used for the preparation of the infants’ food depending on the region, as follows: rural area: tap water 26% (39), boiled 13% (20), and other 3.3% (5); urban area: 50% (75) tap water, boiled 20.6% (31), and other 0.6% (1). The description of the kinds of foods given to the infants is in Table 8. Some infants were fed using a bottle, including 24.6% (37) in the urban area and 12.6% (19) in the rural area. Table 9 shows the water sources for home activities such as dinking, cooking and washing clothes. With regard to the water treatment of drinking water, 48% (72) and 45.3% (68) of cases did not treat drinking water in the rural and urban areas, respectively. The toilet system differed from one household to another, as described in Table 10. It was found that certain households had domestic animals, and the children had contact with them (Table 10).

## 4. Discussion

### 4.1. Socio-Demographic Characteristics

According to the results, male children were more affected by diarrhea than females. This means that the males were more prone to foodborne diarrhea. The finding is in accordance with [15], who found that, in Mozambique, the morbidity and mortality rates in children under five are higher in males than in females. The study shows that the age group with the most cases of foodborne diarrhea were the children under six months from the rural area, compared with the same age group in the urban area, which was unlike in this age group, as the children in this age group were supposed to be exclusively breastfeeding. One should take into account that breastfeeding protects children because it is clean and provides immune supplements to them to prevent diarrhea. However, the safety of exclusive breastfeeding is supported by the good hygiene of the mother. The cases in this age group can be related to the lack of exclusive breastfeeding or poor hygiene of the caretakers. These findings are not in agreement with previous studies. For example, refs. [16,17] found more cases in children above the age of six months. On the other hand, the cases of foodborne diarrhea were concentrated in the groups aged between 12 and 23 months, with more cases in the whole group from the urban area. Adding together both districts, the majority of cases were identified in children under 2 years of age. This may be because children from the age six months start to be introduced to complementary foods, which can cause intolerance and reactions that result in diarrhea. In addition, the children start crawling, walking and making contact with the ground and adopt a tendency to take objects in their mouths. All these conditions can expose the children to the probable agents of diarrhea. This observation is in agreement with previous studies [17,18,19,20,21,22], which reported more cases of diarrhea in children from zero to two years of age.

It was found that the majority of the children were cared for by mothers aged 18 to 32 years. Women at this age are considered as adults who are able to take better care of their children, and they are children’s natural caretakers. The majority of the caregivers were single but living with their partners. Nevertheless, the men were the income providers in the household, and the women were the most likely to be unemployed. Thus, they were supposed to have more time to look after the children. According to [23], children of employed mothers are more susceptive to becoming sick.

Some houses were built with precarious materials, mainly in the rural area, which can be an indicative of the poorest and weakest environmental sanitation that can be a source of the outbreak of diarrhea infections. The finding corroborates with [24], a study conducted in Nigeria. Furthermore, the majority of the households in both regions used charcoal as a fuel for cooking, which can spoil the air and affect the immunity of children, who subsequently become more susceptive to respiratory diseases and more vulnerable to other diseases, including diarrhea. Previous studies also showed the relationship between the type of fuel used in the household and episodes of respiratory diseases in children [25,26,27,28].

### 4.2. Disease-Related Questions

The caregivers in the rural area took the children for medical assistance earlier compared to the urban, and this allowed for quick medical care. The findings show greater concern for children’s health in rural than in urban areas. This fact may be explained by the fact that caregivers in urban areas do not have time to take care of their children, as they work outside the home. This fact is in accordance with [23]. Another study conducted in Pakistan reported that caretakers mostly prefer to take care of children with diarrhea at home in the first 48 h before seeking medical assistance [29]. On the other hand, it was found that the caregivers in the rural area first gave their children traditional medicine before seeking medical assistance. This can put the children at risk, as many traditional medicines have no defined dose and can be given in sub- or overdose quantities and exacerbate the diarrhea, especially if the child is under six months, bearing in mind the fact that the study found more children under six months with diarrhea in the rural area. The findings of this study are similar to the [30] on research performed in Ethiopia.

In all the children with diarrhea, other symptoms were observed, such as vomiting, fever, cough, cold and loss of appetite, among others. Diarrhea and vomiting expose the children to the risk of dehydration and malnutrition, with subsequent death. The symptoms observed in this study may be related to food or water contamination, as some caregivers reported having given spoiled food and unsafe water to the children. In addition, the respiratory symptoms (cough, cold) can be related to environment contamination due to the kinds of fuel used in the households. These findings do not differ from those found in previous studies [16,31,32], associating respiratory symptoms with the use of firewood and charcoal as fuels for cooking meals.

The findings also show that the majority of the children with foodborne diarrhea were not hospitalized. This means that diarrhea infection can be treated easily, without the need for much attention and special care in the hospital. On the other hand, it was observed that the caregivers were concerned with seeking medical assistance as early as possible when the children became ill, and this facilitated the rapid treatment and decreased the need for hospitalization. In this study, more than half of the children were vaccinated for different diseases, including rotavirus, which is considered the main cause of diarrhea infection in children under six months, and this gave them with immunity and minimized the risk of hospitalization due to diarrhea and other infectious diseases. This observation is in agreement with [33,34,35], studies which observed that complete vaccination reduces the risk of death in children under two years, taking into account the fact that rotavirus vaccination is given to children under six months.

In this study, it was also observed that there was direct transmission from one person to another in the same household, where the child became ill before anyone else. Another fact which could increase the spread of diarrhea might be the living conditions, as it was shown that, in rural areas, the house had only one sleeping room. Similar results were reported in previous studies [36,37], reporting that a greater number of individuals in the household is a factor contributing to the increase in diarrheal infection because it compromise the hygiene and sanitation of the household. In addition, [38] emphasized that the presence of caregivers or other persons with diarrhea in the same household can be factor leading to children becoming ill.

According to the caregivers in the rural area, diarrhea in the children was related to spoiled food, traditional beliefs, such as those related to the moon, and teeth growth. Due to these beliefs, the children were treated with traditional remedies, with poor hygiene. Furthermore, in the urban area, there were caregivers who pointed to the consumption of spoiled food and unsafe water as factors associated with diarrhea in the children. The same factors were found in other studies [30,39,40,41]. The place where raw and cooked meat received attention as a probable source of foodborne diarrhea, because in most cases, the local market conditions were not safe, affected by dirt, flies and stagnant water. These findings are in agreement with [42], which observed that the consumption of undercooked meat, especially that from markets with precarious sanitation, could be a health risk.

The type of food given to the infants and the way in which it was given could be other factors affecting the outbreak of foodborne diarrhea. As we observed in the urban region, the children were fed with biscuits, infant formula and pre-prepared food, and they used a bottle washed with untreated water. It is recommended that children under the age of two be fed on semi-solid foods that are easy to digest. Pre-prepared food is not advisable, because the preparation method is not well known, and this can be a possible source of diarrhea in infants. These results are similar with those of other studies [43,44], which reported that incorrect feeding—mainly the introduction of complementary foods—can cause diarrhea in children under two years of age.

With regard to the safety of the water in the urban households, cases of the consumption of unsafe water were observed, even though there is supposed to be clean and safe water in this region. The sources were ponds, which pose a risk of water and foodborne diseases, including diarrhea. The results were similar with those reported in previous studies [45,46,47]. In addition, previous studies reported that, in Mozambique, despite the government’s determination to decrease the numbers of waterborne diseases, there are still many cases of diarrhea due to the ingestion of contaminated water [18,21,46]. A study conducted in Maputo city showed that there is high contamination of fecal microorganisms in the water used for consumption, even if it is piped [48]. It is important to advise the population to treat water before its consumption, especially water used for children’s food and hygiene.

The use of septic tanks in the sanitary systems of the households was found to be positive in terms of diarrhea prevention. However, in the urban area, some families used latrines without lids, which can be a source of flies and other vectors of food and water contamination. This observation corroborates with the studies performed by the authors of [23,49,50]. Furthermore, this study did not enquire about the stage of the tank, but there was a study performed in Maputo which showed that the majority of households reported having never emptied their septic tanks [51]. This can be considered another risk factor for the development of vectors, with the consequent appearance of foodborne diarrhea.

The existence of domestic animals within the households was reported, where most animals were in contact with children and others played with them, principally in rural area. This contact with domestic animals may be a factor affecting diarrhea in children, as the transmission of the disease from animals to human is well known. This finding is in agreement with [52,53], reporting the relationship between animal contact and human being diseases, including diarrhea infection.

## 5. Conclusions

All in all, children under two years of age were more affected by foodborne diarrhea. The mothers were the main caregivers of the children. In the rural area, the children received medical assistance earlier compared to those in the urban area. The diarrhea infections in the rural area were also related to traditional beliefs. Unsafe water, spoiled food, low household income, poor sanitary systems, the type of infant food and the method of preparation, contact with domestic animals, the number of people sleeping in a room were also considered as factors affecting foodborne diarrhea. The study recommends the development of local strategies with caregivers for the prevention and control of foodborne diarrhoea.

## Figures and Tables

**Table 1 ijerph-19-15600-t001:** Genders and ages of caretakers and children.

Gender *n* = 300					Age *n* = 300					
	Caregivers		Children		Caregivers			Children		
	Rural	Urban	Rural	Urban	Interval (Years)	Rural	Urban	Interval (Months)	Rural	Urban
Male	10 (6.7%)	24 (16%)	78 (52%)	88 (58.7%)	18 to 2	39 (25.7%)	41 (27.5%)	<6	35 (23.3%)	25 (16.6%)
					23 to 27	46 (31.1%)	51 (34.2%)	6 to 11	29 (19.1%)	28 (18.6%
					28 to 32	30 (20.9%)	12 (22.1%)	12 to 23	45 (30%)	55 (36.6%)
					33 to 37	16 (10.1%)	12 (8.1%)	24 to 35	19 (12.7%)	18 (12%)
					38 to 42	15 (9.5%)	3 (2.0%)	36 to 47	11 (7.3%)	9 (6%)
Female	140 (93.3%)	125 (84%)	72 (48%)	62 (41.3%)	43 to 47	3 (2.0%)	2 (1.3%)	48 to 59	11 (7.3%)	15 (10%)
					48 to 52	0	1 (0.7%)			
					53 to 57	1 (0.7%)	4 (2.7%)			
					>63	0	2 (1.3%)			
Total	150	150	150	150		150			150	150

**Table 2 ijerph-19-15600-t002:** Caregivers’ occupations and incomes.

Occupation			Monthly Income, Metical		
	Rural	Urban		Urban	Rural
Housewife	20 (13%)	29 (19%)	<4000	4 (3%)	3 (2%)
Unemployed	88 (59%)	78 (52%)	4000 to 8000	6 (4%)	9 (6%)
Merchant	13 (9%)	12 (8%)	8000 to 12,000	2 (1%)	3 (1%)
Outdoor manual worker	6 (4%)	7 (5%)	12,000 to 16,000	1 (1%)	0
Guard/police officer	6 (4%)	7 (5%)	16,000 to 20,000	2 (1%)	
			20,000 to 24,000	0	0
Driver	5 (3%)	6 (4%)	24,000 to 28,000	0	0
Gardener	5 (3%	5 (3%)	28,000 to 32,000	0	0
Teacher	4 (3%)	3 (2%)	32,000 to 36,000	1 (1%)	0
Soldier	3 (2%)	3 (2%)	>36,000	1 (1%)	0
			Do not Know	133	135
Total	150	150		100	100

**Table 3 ijerph-19-15600-t003:** Number of rooms and average number of people living in the household.

	Rooms			Average Number of People Living in the Household	
	Rural	Urban		Rural	Urban
1	59 (39%)	37 (25%)	2 to 3	48 (32%)	26 (17%)
2	51 (34%)	24 (16%)	4 to 5	61 (41%)	50 (33%)
3	28 (19%)	41 (27%)	>6	41 (27%)	74 (49%)
>4	12 (8%)	48 (32%)			
Total	150	150		150	150

**Table 4 ijerph-19-15600-t004:** Building materials and fuel used for cooking.

Floor Material			Roof Material			Wall Material			Fuel for Cooking		
	Urban	Rural		Urban	Rural		Urban	Rural		Urban	Rural
Cement	93 (62%)	79 (53%)	Iron plate	102 (68%)	110 (73%)	Cement block	108 (72%)	98 (65%)	Charcoal	45 (30%)	49 (33%)
Ceramic tiles	54 (36%)	53 (35%)	Cement	22 (15%)	29 (19%)	Straw	20 (13%)	40 (27%)	Coal	35 (23%)	14 (9%)
Sand	3 (2%)	17 (11%)	Melamine	23 (15%)	10 (%)	bricks	22 (15%)	12 (8%)	Cooking gas	33 (22%)	31 (21%)
Others	0	1	Others	3 (2%)	1 (1%)				Electricity	25 (17%)	18 (12%)
									Wood	9 (6%)	32 (21%)
									Others	3(2%)	6(4%)
Total	150	150		150	150		150	150		150	150

**Table 5 ijerph-19-15600-t005:** Events before receiving medical assistance.

Days before Medical Assistance			Home Treatment			Signs and Symptoms		
	Rural	Urban		Rural	Urban		Rural	Urban
0 to 3	122 (81%)	57 (38%)	Antibiotic	0	2 (1%)	Diarrhea	150	150
4 to 6	6 (4%)	4 (3%)	Do not know	77 (51%)	141 (94%)	Vomit	28 (19%)	37 (25%)
>7	13 (9%)	6 (4%)	Nothing	47 (31%)	4 (3%)	Fever	18 (12%)	22 (15%)
Do not know	9 (6%)	83 (55%)	ORS (oral rehydration solution)	1 (1%)	0	Cough	25 (17%)	24 (16%)
			Paracetamol	2 (1%)	0	Cold	24 (16%)	15 (10%)
			Traditional medicine	22 (15%)	3 (2%)	Loss of appetite	28 (19%)	30 (20%)
			Water	1 (1%)	0	Abdominal pain	12 (8%)	6 (4%)
						Bright stools	5 (3%)	2 (%)
						Others	1 (1%)	3 (2%)
						Headache	7 (%%)	3 (2%)
						Weakness	1 (1%)	2 (1%)
						Dizziness	0	1 (1%)
						Obstipation	0	1 (1%)
						Fatigue	1 (1%)	2 (1%)
						Joint pain	0	1 (1%)
						Dysuria	0	1 (1%)
Total	150	150		150	150		150	150

**Table 6 ijerph-19-15600-t006:** Possible sources of diarrhea.

Source		
	Rural	Urban
Cloudy water	4 (3%)	10 (7%)
Sandwich	5 (3%)	1 (1%)
Fermented food	7 (5%)	6 (4%)
Teething	23 (15%)	8 (5%)
Artificial milk	5 (3%)	0
Moon	11 (7%)	6 (4%)
Mango	9 (6%)	1 (1%)
I do not know	47 (31%)	105 (70%)
Others in small frequency (0–4)	39 (26%)	13 (9%)
Total	150	150

**Table 7 ijerph-19-15600-t007:** Places of food purchase.

	Raw Meat		Cooked Meat	
Purchase Place	Rural	Urban	Rural	Urban
Local Market	113 (75%)	57 (65%)	32 (21%)	14 (9%)
Supermarket	26 (17%)	23 (15%)	2 (1%)	5 (3%)
Butcher	8 (5%)	6 (4%)	8 (5%)	2 (1%)
Others	2 (1%)	5 (3%)	6 (4%)	0
Not applicable (do not eat meat)	1 (1%)	13 (9%)	6 (4%)	14 (9%)
Directly from the producer	0	0	0	1 (1%)
Self-sufficient	0	0	0	0
No answer			96 (64%)	114 (76%)
Total	150	150	150	150

**Table 8 ijerph-19-15600-t008:** Types of food given to the infant.

Infant Food	Urban	Rural
Biscuits	48 (44%)	37 (34%)
Fermented cereals	3 (3%)	18 (17%)
Breastfeeding	10 (9%)	19 (17%)
Fruit or vegetable juices	5 (5%)	4 (4%)
Infant formula	18 (17%)	14 (23%)
Pre-prepared meals	10 (9%)	7 (6%)
Fruit puree	3 (3%)	2 (2%)
Other foods	2 (2%)	4 (4%)
Formula for < 6 months	9 (8%)	4 (4%)
Total Number of Infants	108	109

**Table 9 ijerph-19-15600-t009:** Water sources for house activities.

	Drinking		Washing		Food Preparation	
Type of Water	Urban	Rural	Urban	Rural	Urban	Rural
Tap water	116 (73%)	138 (92%)	116 (73%)	138 (92%)	116 (73%)	138 (92%)
Public/community well or pump	22 (15%)	12 (8%)	22 (15%)	12 (8%)	22 (15%)	12 (8%)
Bottled water	5 (3%)	0	5 (3%)		5 (3%)	
Private well or pump with lid	4 (3%)	0	4 (3%)		4 (3%)	
Pond	1 (1%)	0	1 (1%)		1 (1%)	
Other	1 (1%)	0	1 (1%)		1 (1%)	
Tank	1 (1%)	0	1 (1%)		1 (1%)	
Total	150	150	150	150	150	150

**Table 10 ijerph-19-15600-t010:** Type of sanitary equipment in the household and animals in the household.

Toilet System			Animal in Household			Child Had Direct Contact with Animals	
	Urban	Rural		Rural	Urban	Rural	Urban
Toilet with septic tank	85 (57%)	138 (92%)	Chickens	20 (13%)	26 (17%)	6 (4%)	6 (4%)
Latrine with roof protection	30 (20%)	9 (6%)	Ducks	10 (7%)	13 (9%)	7 (5%)	7 (5%)
Improved latrine with ventilation	28 (19%)	2 (1%)	Pigs	2 (1%)	2 (1%)	1 (1%)	0
Latrine without roof protection	4 (3%)	1 (1%)	Cats	15 (10%)	13 (9%)	11 (7%)	11 (11%)
Others	2 (1%)	0	Dogs	8 (5%)	8 (5%)	4 (3%)	2 (1%)
No sanitary system	1 (1%)	0	Other	20 (13%)	2 (1%)		
			No animals	75 (50%)	86 (57%)	121 (80%)	126 (84%)
Total	150	150		150	150	150	150

## Data Availability

Not applicable.

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
