# Peer review of "Household Factors of Foodborne Diarrhea in Children under Five in Two Districts of Maputo, Mozambique"

_ijerph, 2022, doi:10.3390/ijerph192315600_

Round 1

Reviewer 1 Report

Please could you add more statistical analysis please such as p-value or Odds ratio regarding ptoential risk factor?

Author Response

Dear reviewer

Thank you for your comment, please find attech the response.  

Reviewer 2 Report

The paper presents the results of a comprehensive study on household factors that contribute to the development of diarrhea in children in Mozambique. The work is based on data obtained from a structured questionnaire that is not attached to the manuscript. The study is extensive and detailed, but presented in a rather confusing way that distracts the reader.

Some of the specific comments are:

It is not clearly explained what urban and rural areas mean.

It is not stated how many participants refer to those from urban or rural areas (are both groups equally represented?)

It was stated that in this study more than half of the children were vaccinated, but it is not clear against which disease. Is it a rota virus that is a potential cause of diarrhea in children?

The tables are unclear to the reader

In the discussion, the similarities or differences of the obtained results with the data from the literature should be briefly described, and not only indicated by the reference number ("This finding is in agreement with 52, 53)

Author Response

Dear Reviewer 

Thank you for your comments, please find attach the response 

Reviewer 3 Report

The article "Household factors of foodborne diarrhoea in children under five years of age in two districts of Maputo- Mozambique" provides relevant information on the factors contributing to diarrhoea in children under five years of age in Mozambique. However, these factors are already known, such as water quality, personal hygiene and waste management. Moreover, this study does not emphasize discussing or explaining the difference between rural and urban areas, as well as, in other results, proposing measures to minimize the ill risk. Furthermore, the manuscript should be improved in its organization and consistency concerning writing text.

Moreover,  I would like to include specific questions or comments:

·         The study was performed in two districts. How many inhabitants do they have? How many of them are children under 5 years of age? And how many of them belong to rural and urban areas?

·         In Table 1:  decimals commas should be replaced by decimal points.

·         In Table 2:  there are two rows with unemployed; what is the difference between them?

·         In Table 3, the term Persons per household are the overall number of respondents. Perhaps interesting data would be the average number of people living in the households, and it could be a factor that could affect the spread of disease.

·         Table 5. ORS, meaning oral rehydration solution?

·         Line 129-130. In the sentence "According to the result male children were more affected by diarrhoea than female. This meaning that the males are more prone to foodborne diarrhoea. The finding is in accordance with [15]." The author should be included. For example: "According to the result, male children were more affected by diarrhoea than female. This meaning that the males are more prone to foodborne diarrhoea. The finding is in accordance with UNICEF (2021)". This remark is repeated elsewhere in the manuscript.

·         Is there any explanation for why male children are more affected by diarrhoea?

Author Response

Dear Reviewer

Thank you for your comments, please find in attach the response

Round 2

Reviewer 2 Report

After the corrections made, the manuscript is more comprehensible even for those who do not have expertise in this area.

Reviewer 3 Report

The manuscript has been substantially improved after responding adequately to the suggestions derived from the revision process

However, I would like to make a small remark about the bibliography in the text, especially in the discussion section.

For example:

Line 171: The finding is in accordance with [15] found that in Mozambique…. It should be included the author/s…” The finding is in accordance with UNICEF (2021), which reported that in Mozambique....

 Other examples:

Line 195: According to [23] children of employed mothers are  more susceptive to be sick. It should be replaced by “According to Bashuru (2019) children of employed mothers are  more susceptive to be sick”.

See too: Line 241, 249, 251, 262, 290….